# Design Motifs for Probabilistic Generative Design

**Geoffrey Roeder,**[*] **Nathan Killoran,**[*] **Will Grathwohl & David Duvenaud**
Department of Computer Science
Vector Institute for Artificial Intelligence, University of Toronto
{roeder, wgrathwohl, duvenaud}@cs.toronto.edu, nkilloran@psi.toronto.edu

## Abstract

Generative models can be used to produce designs that obey hard-to-specify constraints while still producing plausible examples. Recent examples of this include drug design, text with desired sentiment, or images with desired captions. However, most previous applications of generative models to design are based on bespoke, ad-hoc procedures. We give a unifying treatment of generative design based on probabilistic generative models. Some of these models can be trained end-to-end, can take advantage of both labelled and unlabelled examples, and automatically trade off between different design goals.

## 1 Introduction

The goal of computer-aided design is to automate parts of the creative process. We would like to be able to specify some desired properties, and have an algorithm produce a diverse set of suitable candidates. By having an algorithm perform some of the rudimentary design tasks, a human designer would be freed to take on a more high-level creative role in the design process.

Recently, Variational Autoencoders (VAEs) Kingma & Welling (2014); Rezende et al. (2014), Generative Adversarial Networks (GANs) Goodfellow et al. (2014), deep autoregressive networks van den Oord et al. (2016b;a), have achieved impressive performance at simulating highly-structured real-world data. Other recent work in model architecture and search such as Mueller et al. (2017) and Engel et al. (2017) has also produced impressive results. However, it remains an active area of research as to how best approach the problem of learning optimal representations for the task of automated design of highly structured data.

We propose a general, domain- and algorithm-agnostic formulation of the task of probabilistic generative design, namely as sampling from a probability distribution conditioned on relevant targets and constraints. We present a framework for an exhaustive taxonomy of different graphical models that allow us to perform automated design in a probabilistic setting, contrasting models that explicitly model latent variables with those that directly optimize a data-generating process.

We empirically investigate the geometric properties of generative design models, showing that the set of relevant designs (the "design manifold") can be complex, disconnected, and have low volume relative to the larger design space. We show that explicitly modeling latent variables and training using semi-supervision can be used to align the design manifold, thus making the design task easier.

## 2 Automated Design through Probabilistic Generative Models

How can we formulate automated design of structured data as a probabilistic generative modeling and inference task? Our underpinning assumption is the manifold hypothesis (Cayton, 2005; Narayanan & Mitter, 2010; Brahma et al., 2016): real-world data forms a distribution $p(\mathbf{x})$ which is concentrated near a low-dimensional manifold embedded in a larger space. We will refer to this as the *data manifold*. For example, in an action-recognition dataset, the manifold hypothesis states that all the information about the actions could be represented by the lower-dimensional sub-manifold of the joint locations of the individuals.

---

[*]Equal contribution

VAEs and GANs have largely solved the problem of producing realistic data from the data manifold, but not in a targeted manner in the absence of post-hoc methods. Hence, the key criteria for automated design from a model of data is the ability to impose design constraints or target attributes. Such constraints allow us to limit generations from the model to sub-regions of the manifold. We denote such target attributes with a composite *target variable* $\mathbf{t}$, which specifies the desired attributes.

Conditioning on a target value turns the distribution $p(\mathbf{x})$ into a conditional distribution,

$$p(\mathbf{x}|\mathbf{t}) \propto p(\mathbf{x})p(\mathbf{t}|\mathbf{x}).$$

When the conditional distribution is concentrated near a manifold, we call it a *design manifold*. Note that sampling from $p(\mathbf{x}|\mathbf{t})$ automatically achieves the goals of generative design:

- **Plausibility** is automatically enforced by $p(\mathbf{x})$, which might be defined implicitly by a VAE or GAN. This model can be trained from unlabeled data.
- **Targeted** The term $p(\mathbf{t}|\mathbf{x})$ automatically enforces that that samples from $p(\mathbf{x}|\mathbf{t})$ necessarily be a subset of the possible unconditional plausible configurations.
- **Diversity** A well-calibrated model will automatically put mass on all valid possibilities. When training a VAE, we can explicitly see that attempting to fit a good model introduces a term promoting entropy.

Indeed, probabilistic generative models support many of these goals through inference. In particular, inference gives us: a baked-in **measure for quality**, since we can assess fidelity to the label distribution through $p(\mathbf{t}|\mathbf{x})$; **efficient search**, since we can use gradient-based optimization on the latent representation; and **automatic model choice**, because a Bayesian approach selects model capacity and regularization parameters during inference.

## 3 PROBABILISTIC GENERATIVE DESIGN (PGD) MOTIFS

In this section, we preview a comprehensive taxonomy of graphical model motifs. The motifs presented are four of a large possible number of directed graphs involving $\mathbf{x}, \mathbf{t}, \mathbf{z}$. They exemplify the interaction between model choice, implicit or explicit latent representations, and the target functions being optimized. We leave a full treatment of such motifs, such as considerations of sampling and inference, to future work. Here, we show how adopting a PGD framework reveals useful insights into automated design in different practical contexts. We explore one such insight empirically with a semi-supervised VAE on MNIST to exhibit the strengths of the common cause motif.

### 3.1 SELECTED NON-LATENT VARIABLE MOTIFS

**Direct Generation Conditional on Target** The ideal model would be one that could directly sample $\mathbf{x} \sim p(\mathbf{x}|\mathbf{t})$. For example, a neural network could be trained to directly generate an image given a caption, an approach taken with mixed results in Mansimov et al. (2015). If $\mathbf{t}$ can only take a few values, and data is plentiful, training such a model is straightforward, as in Dosovitskiy et al. (2017). However, if $\mathbf{t}$ lives in an open-ended space of possible design criteria, such as the set of all possible captions, then training such a model is difficult, since it requires learning a mapping from one high-dimensional space to another. 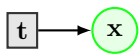

**Regression and Search** The standard approach to machine-learning based design without the use of generative models is based on regression and search. In this pattern, one learns a function mapping $f : \mathcal{X} \to \mathcal{T}$ from labeled examples, then performs optimization to find an $\mathbf{x}$ that approximately maximizes $f(\mathbf{x})$. If $\mathbf{x}$ and $\mathbf{t}$ are continuous then gradient-based optimization can be used, as in Mordvintsev et al. (2015). However, this approach has two major drawbacks: 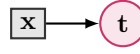

- Without a generative model $p(\mathbf{x})$ to constrain the design, the samples $\mathbf{x}$ might veer wildly away from typical data. Thus, the search space must be carefully chosen to contain only valid designs.
- In discrete settings, optimization is difficult due to the lack of gradients to inform the search direction.

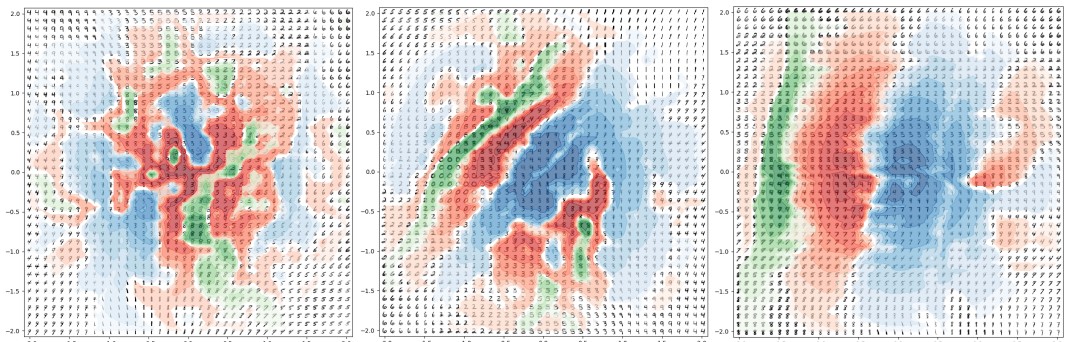

Figure 1: Effect of increasing proportion of labelled training data for a semi-supervised VAE. Colors are densities over regions of high-target-value reconstructions: green is highest, blue is lowest. A high target value reconstruction is one with many active (black, binary-valued) pixels.

## 3.2 SELECTED LATENT VARIABLE MOTIFS

**Plug and Play**   Nguyen et al. (2016a;b) addressed one of the main problems with the regression-and-search approach. They constrained the search space of possible images to the subset of natural images by coupling a classifier (mapping $f : \mathcal{X} \to \mathcal{T}$) with a generative model mapping from an unconstrained latent space to the low-dimensional manifold of natural images ($\mathcal{Z} \to \mathcal{X}$). However, this approach still has several drawbacks:

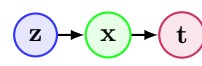

- Evaluating the gradient $\frac{\delta \mathbf{t}}{\delta \mathbf{z}} = \frac{\delta \mathbf{t}}{\delta \mathbf{x}} \times \frac{\delta \mathbf{x}}{\delta \mathbf{z}}$ requires differentiating through both a high-resolution generative model, as well as a high-resolution discriminative model.
- For high dimensional $\mathbf{x}$, training such a model can require an excessive amount of labels.

**Common Cause**   This approach was explored by Gómez-Bombarelli et al. (2016). This model has the advantage that it can be used to optimize discrete designs.

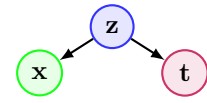

- Training the target function from $\mathbf{z} \to \mathbf{t}$ requires encoding the corresponding $\mathbf{x}$ values. This is straightforward if the generative model is a VAE, but might not always be available if using a GAN.
- Joint training with labels can exploit gradient information to organize the latent space (see fig. 1)

## 3.3 SEMI-SUPERVISED TRAINING ORGANIZES DESIGN MANIFOLDS

Figure 1 shows that design manifolds are disconnected and nonlinear in the absence of supervision. Training with semi-supervision organizes regions of the latent space. The data are binarized MNIST digits, and the labels correspond to the number of active pixels in the data point. The models employ the "common cause" motif: each subfigure represents a semi-supervised VAE model, trained to convergence with a fixed amount of labelled data (increasing towards the right). See Appendix A for details of loss function derivation and training.

## 4 CONCLUSIONS

Focusing on even a few design motifs exposed useful insights for future work in automated design. Out of all the motifs we considered, the common cause model stands alone in both allowing semi-supervised learning, gradient-based inference and search, and requiring learning only low-dimensional functions. Future research will also consider interactions between approximate inference strategies, and the form of the models chosen.

We believe the language of design motifs captures essential, model agnostic information relevant to the use of probabilistic generative models for automatic design. Ongoing work in this area will expand the lexicon of design motifs to explore strengths and weaknesses of each motif, and develop knowledge of inference and sampling characteristics to support rapid progress in probabilistic generative design.

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

## A    Semi-supervised Variational Autoencoder (SSVAE): Loss and Training

The images in figure 1 exhibit three conditional latent density functions superimposed on each trained SSVAE, where the regions of high and low density are visualized over a grid of reconstructions in the latent space. In this section, we explain the model and training of these latent densities in detail. Code for declaring and training the model may be found at www.github.com/geoffroeder/design-motifs.

### A.1    Training

MNIST digits are modelled at the pixel level as i.i.d. Bernoulli random variables. For approximate posterior distributions, we use a diagonal Gaussian. We define a target function $t(\mathbf{x}) = \sum_{j \in |G(\mathbf{z})|} G(\mathbf{z})_j$, where $\mathbf{x} = G(\mathbf{z})$, e.g., the reconstruction of some point in the latent space. This distribution captures the number of active pixels of the datapoint. This function correlates well with the thickness of the digit, as observed in figure 1. To exhibit how simple modelling choices can still yield generative models useful for design, we model $p(\mathbf{t}|\mathbf{z})$ as linear regression, e.g., a Gaussian also. The same latent space is trained for regeneration of data and labels through a simple semi-supervised VAE (SSVAE).

We train this model by forming a common cause evidence lower bound, noting first that since each observation from the data distribution is i.i.d. by assumption, we may factorize these observations into those with an associated label $t$ and those without a label:

$$\log p(\{\mathbf{x}_u\}, \{\mathbf{x}_L, \mathbf{t}\}|\theta_x, \theta_t) = \log \left( \prod_{i=1}^{N_u} p(\mathbf{x}_i) \prod_{j=1}^{N_l} p(\mathbf{x}_j, \mathbf{t}_j) \right)$$

$$= \sum_{i=1}^{N_u} \log p(\mathbf{x}_i) + \sum_{j=1}^{N_l} \log p(\mathbf{x}_j, \mathbf{t}_j),$$

where $N_u$ is the number of unlabelled datatpoints and $N_l$ the number of labelled datapoints. We then introduce a latent space $\mathcal{Z}$, and two differentiably reparameterizable variational families $q$ parameterized by $\phi_x$ and $\phi_t$ to approximate the intractable posterior distributions:

$$\sum_{i=1}^{N_u} \log \int p(\mathbf{x}_i|\mathbf{z})p(\mathbf{z})d\mathbf{z} + \sum_{j=1}^{N_l} \log \int p_{\theta_x}(\mathbf{x}_j|\mathbf{z})p_{\theta_t}(\mathbf{t}_j|\mathbf{z})p(\mathbf{z})d\mathbf{z}$$

$$\geq \sum_{i=1}^{N_u} \mathbb{E}_{q_{\phi_x}} \left[ \log p_{\theta_x}(\mathbf{x}_i, \mathbf{z}) - \log q_{\phi_x}(\mathbf{z}|\mathbf{x}_i) \right]$$

$$+ \sum_{j=1}^{N_l} \mathbb{E}_{q_{\phi_t}} \left[ \log p_{\theta_x}(\mathbf{x}_j, \mathbf{z}) + \log p_{\theta_t}(\mathbf{t}_j|\mathbf{z}) - \log q_{\phi_t}(\mathbf{z}|\mathbf{x}_j, \mathbf{t}_j) \right].$$

By mapping and regenerating both digits and labels from the same latent space, the common cause model provides a simple form of scaffolding for this space that encourages it to pack information about the label along the latent dimensions, as demonstrated in figure 1.

### A.2    Visualization

We model the data using the common cause motif. This allows us to take advantage of factorization of the joint distribution for visualization purposes. For a particular reconstruction mean $\mathbf{z}$ of any datapoint $\mathbf{x}$, we can use Monte Carlo sampling to approximate the conditional density of $p(\mathbf{z}|\mathbf{t})$ at

point. Specifically,

$$
\begin{aligned}
p(\mathbf{z}|\mathbf{t}) \propto p(\mathbf{z}, \mathbf{t}) &= \int p(\mathbf{z}, \mathbf{t}, \mathbf{x}) d\mathbf{x} \\
&= \int p(z) p(\mathbf{x}|\mathbf{z}) p(\mathbf{t}|\mathbf{z}) d\mathbf{x} \\
&= \mathbb{E}_{p(\mathbf{x}|\mathbf{z})} \left[ p(\mathbf{z}) \mathbb{I} \left[ t(\mathbf{x}|\mathbf{z}) \in \mathcal{V} \right] \right] \\
&\approx p(\mathbf{z}) \frac{1}{S} \sum_{i=1}^{S} \mathbb{I} \left[ t(\mathbf{x}_i|\mathbf{z}) \in \mathcal{V} \right].
\end{aligned}
$$

We divide the empirical output range of true labels $t$ calculated over MNIST into three roughly equal-sized ordered sets $\mathcal{V}_{1:3}$. The density for the set containing the smallest values is coloured blue, the middle region of values red, and the greatest region of values green.

