# OpenReview forum: "Design Motifs for Probabilistic Generative Design"
_ICLR.cc/2018/Workshop — Reject_

### Official Review · AnonReviewer2 · 2018-03-11
**interesting taxonomy**

**Rating:** 7
**Confidence:** 3

**Review:**

I enjoyed the author's approach for design motifs or patterns. I think these kinds of papers are sorely absent from existing machine learning literature, and are useful for spurring discussion, so I encourage acceptance of this paper. The writing can be cleaned up a bit; the concepts and definitions should be made more precise before acceptance.

Slight nitpicky comments:

* I was confused about what automated design means. Please remove this language or define it. (Specifically, you say the human designer would be freed to take on higher-level, but still "design", work.)

* I'm confused about what the constraints are. The $$ \mathbb{t} $$ variable is central to your taxonomy, but it is ill-defined. Please give several examples of why your definition of t captures common applications (drug design, images, etc), and what it *doesn't* capture.

* I'm confused by this: "When training a VAE, we can explicitly see that attempting to fit a good model introduces a term promoting entropy". This is confusing: 'attempting to fit a good model' should say 'choosing to use variational inference to fit an approximate posterior distribution introduces a term promoting entropy of the learned approximate posterior over latent variables'. I know this is wordy, but it hopefully promotes better taxonomy than the sentence.

---

### Official Review · AnonReviewer1 · 2018-03-12
**not up to par for publication at iclr workshop**

**Rating:** 4
**Confidence:** 4

**Review:**

Summary:
The paper's claimed contributions are:
1- propose a taxonomy of graphical models for generative design
2- perform empirical study of the geometric properties of design models. Concluded that use of latent variables and semi-supervised training are good for generative design modeling.

Pros:
Conditional generation is an important task to study

Cons:
--feels like a short review paper rather than an innovative idea
--for the common cause motif: if you condition on z (which you are doing because you shaded z) then x and t are independent from each other. This seems to be contradictory from the initial desire to generate x from p(x | t)!
--section 3.3 is somewhat disconnected from the previous sections. I understand you only have three pages but I would have favored an even shorter intro for a clearer exposure

Minor:
--please use \citep in the introduction. your citation style in the introduction is distracting
--it is vague to say "support many of these goals through inference" when there are only three goals

---

### Decision · Program_Chairs · 2018-03-20
**ICLR 2018 Workshop Acceptance Decision**

**Decision:**

Reject

**Comment:**

Based on the reviews, this paper has not been accepted for presentation at the ICLR workshop. However, the conversation and updates can continue to appear here on OpenReview.